# Preparation of ZnO Nanosheet Array and Research on ZnO/PANI/ZnO Ultraviolet Photodetector

**DOI:** 10.3390/polym15224399

**Published:** 2023-11-14

**Authors:** Xuanzhen Zhang, Yunhui Feng, Fangbao Fu, Huan Wang

**Affiliations:** 1Guangzhou Key Laboratory of Sensing Materials & Devices, Center for Interdisciplinary Health Management Studies, College of Physical Education, School of Chemistry and Chemical Engineering, Guangzhou University, Guangzhou 510006, China; zxz@e.gzhu.edu.cn; 2Guangdong Provincial Key Laboratory of Plant Resources Biorefinery, School of Chemical Engineering and Light Industry, Guangdong University of Technology, Guangzhou 510006, China; fufangbao@gdut.edu.cn

**Keywords:** ZnO nanosheets, PANI nano-porous film, ultraviolet photodetector

## Abstract

High-performance ultraviolet photodetectors have important scientific research significance and practical application value, which has been the focus of researchers. In this work, we have constructed a highly photosensitive UV photodetector with a unique “sandwich” structure, which was mainly composed of two layers of ZnO nanosheet arrays and one layer of polyaniline (PANI). The results showed that the UV current of ZnO/PANI devices was 100 times higher than that of pure ZnO devices under the same UV irradiation time. At a 365 nm wavelength, the device had excellent photocurrent responsiveness and photoconductivity. This high performance was attributed to the large specific surface area of ZnO nanosheets and the p-n junction formed between P-type PANI nano-porous film and N-type ZnO nanosheets. This provides a solid theoretical basis for the application of ZnO nanosheets in ultraviolet detection, and possesses significance for the development of ultraviolet photodetectors.

## 1. Introduction

Ultraviolet (UV) radiation, with a wavelength of 10–400 nm, is one of the strongest radiations in nature and has a profound impact on the survival and evolution of life on the Earth. Therefore, the increasing demand for ultraviolet radiation monitoring in many fields has promoted research into various ultraviolet photodetectors [1]. Especially in recent years, ultraviolet photoelectric detectors have been widely used in communication detection [2], aerospace [3], medicine [4], biology [5] and other fields. As is well known, traditional ultraviolet photodetectors typically require the application of an external bias voltage to separate photogenerated charge carriers and enhance their photoresponse characteristics. p-n junctions, which transform ultraviolet radiation into electrical signals through photovoltaic effects, are the foundation of broadband gap oxide semiconductor UV photodetectors. Compared with traditional bulk materials or thin films, UV photodetectors built on semiconductor materials, particularly one-dimensional nanostructures like nanoribbons, nanotubes and nanowires, typically have highly competitive performance advantages due to their volumetric light absorption. And these devices have the advantages of lower power consumption, more functions, and higher responsiveness and accuracy [6,7,8,9,10].

Nanostructured oxide semiconductor materials exhibit superior and unique physical properties due to their large surface-area-to-volume ratio and quantum confinement effects [11,12,13]. As a result of their outstanding device performance, one-dimensional photodetectors made of ZnO wide bandgap nano oxide semiconductor materials have attracted much research [14,15,16,17,18]. ZnO is a wide bandgap semiconductor material with a direct bandgap of 3.37 eV [19,20] and an exciton binding energy of up to 60 meV [21]. It has good thermal and chemical stability, and is low-cost and easy to obtain. It has been identified as one of the most effective antireflective semiconductor materials for optical detecting applications [22,23] and is widely used in various fields, including UV ultraviolet optoelectronic devices [24], solar cells [25] and light-emitting diodes [26]. However, ZnO is susceptible to corrosion by light, resulting in low visible light utilization efficiency, high recombination rate of photogenerated electron hole pairs and low quantum efficiency, and it is challenging to obtain stable performance due to the inherent self-compensation effect of its donor. Because of this, creating high-efficiency UV photodetectors based on ZnO homojunction is almost impossible [27,28,29]. Compared with bulk structures, the nanostructured form of ZnO provides excellent optical and electrical properties, and the composite modification of nanostructured ZnO with semiconductors can reduce the recombination rate of photogenerated electron hole pairs, which improves the quantum efficiency [30].

This has aroused widespread interest among researchers based on the characteristics and advantages of ZnO. Huang [31] prepared UV photodetectors based on ZnO nanorods and Al nanoparticles. According to the research, adding Al nanoparticles greatly improved the photoelectric performance of ZnO-based UV photodetectors. The mechanism is that Al nanoparticles are irradiated on ZnO nanorods, which enhances absorption and causes LSPR to take place. Grigoryev, LV et al. [32] reported the optical photoluminescence and photovoltaic properties of ZnO-LiNbO_3_ thin film. The X-ray structural analysis results of ZnO films synthesized on single-crystal lithium niobate substrate and quartz glass substrate were introduced. The transmittance, reflectance, absorption, photoluminescence and photocurrent spectra of ZnO-LiNbO_3_ thin film and ZnO-SiO_2_ structure in the ultraviolet and visible spectral regions were given. Yin et al. [33] created Cu and Ag co-doped ZnO nanorod arrays on p-GaN/Al_2_O_3_ substrates using a low-temperature hydrothermal method, and investigated the effects of co-doping on morphology, microstructure and electrical/optical properties. Compared with other samples, the average diameter of co-doped ZnO NRs is larger and the unit area (density) of NRs was lower. Ag^+^Cu co-doping can also reduce the bandgap of ZnO NRs, which can greatly improve the electrical performance of heterojunctions, demonstrating the potential application of Ag^+^Cu co-doping ZnO NRs in UV-light-emitting diodes.

In conductive polymers, the high conductivity of the polymers can promote rapid charge transfer, thereby improving the overall performance of the device. PANI is used as a conductive material in ZnO nanosheets for composite modification due to its unique redox properties. And those can be modified to meet specific requirements by changing synthesis conditions or doping PANI with various components, as well as significant electrochemical performance, environmental stability, excellent conductivity, high theoretical capacitance and low cost [34,35,36]. Muhammad Naveed ur Rehman et al. [37] prepared raw ZnO, Y_2_O_3_, binary PANI-Y_2_O_3_, PANI-ZnO, Y_2_O_3_-ZnO and novel ternary PANI-Y_2_O_3_-ZnO nanocomposites using co-precipitation and ultrasound techniques in this study. It has been established that the highest specific capacitance value of the novel ternary nanocomposite material was due to the fast charge transfer rate and enhanced surface dependent electrochemical performance of PANI.

However, for a long time, the preparation process of ZnO nanosheets was complicated, and the experimental reproducibility and sample uniformity were not high, which became an obstacle to the application of ZnO nanosheets in optoelectronic components [38]. In recent years, a number of different techniques for the synthesis of ZnO nanostructures have been reported both domestically and internationally, such as electrodeposition, chemical oxidation and chemical bath deposition. Electrochemical deposition, known as electrodeposition or electroplating, is a popular chemical method for growing crystalline seed layers. It can improve deposition quality and enhance adhesion, which greatly helps in the synthesis of ZnO nanostructures with different structures. Due to the fact that electrochemical deposition only permits the growth of nanostructures when voltage is applied, that can effectively reduce the loss of chemical materials, save resources and eliminate waste, which is in line with the national sustainable development strategy. Other advantages of the electrochemical deposition method include easy stoichiometric control by adjusting the deposition voltage, the ability to control the deposition rate by adjusting the applied current density, ease of doping semiconductors, and the possibility of designing or manipulating the bandgap of nanomaterials [39].

In this work, uniform ZnO nanoparticle arrays were grown directly on the surface of ITO conductive glass by means of electrodeposition and chemical oxidation [40], eliminating the complicated process of preparing the traditional ZnO seed layer. A unique “sandwich” structure UV photoelectric detection device based on ZnO nanosheet arrays was designed and prepared, and a series of photoelectric performance tests were carried out. The results showed that the UV current of ZnO/PANI devices was 100 times higher than that of pure ZnO devices under the same UV irradiation time. At a 365 nm wavelength, the device had excellent photocurrent responsiveness and photoconductivity.

## 2. Materials and Methods

### 2.1. Chemicals

Zinc nitrate hexahydrate (Zn(NO_3_)_2_·6H_2_O, 99.99%) was purchased from Aladdin (Shanghai, China). Potassium chloride (KCl) was purchased from Tianjin Kemeier Chemical Reagents Factory (Tianjin, China). Aniline (C_6_H_7_N, 99%) and ammonium peroxosulfate (APS, 98%) were purchased from Alfa Aesar (Shanghai, China). Hydrochloric acid was purchased from Guangzhou Second Chemical Reagent Factory (Guangzhou, China). Absolute ethanol was purchased from Tianjin Yongda Chemical Reagent Co., Ltd. (Tianjin, China). ITO substrates were purchased from Nippon Plate Glass Co., Ltd. (Kusatsu, Japan). All chemicals were of analytical grade and used as received without further purification.

### 2.2. Preparation of ZnO NSs Array

The ZnO NSs array was electrodeposited on the indium-doped tin oxide (ITO) glass substrates using an easier procedure without preparing a seed layer on it. Firstly, the ITO substrates were ultrasonically cleaned for 20 min in acetone, ethanol and deionized water. Then, they were dried and placed in a stainless-steel electrode clamp. The arrays of ZnO NSs were electrodeposited on the ITO substrates in the aqueous electrolyte of 0.05 mol/L Zn(NO_3_)_2_ and 0.1 mol/L KCl with a three-electrode electrochemical system. In the three-electrode system, a platinum sheet was used as the counter electrode, ITO as the working electrode and the saturated calomel electrode (SCE) as the reference electrode. The conductive surface of ITO conductive glass should be parallel to the platinum electrode. Electrodeposition was then performed at −1.1 V, 50 °C for 30 min using an electrochemical workstation. Finally, the ZnO NSs arrays were obtained after being rinsed with deionized water and dried in ambient air.

### 2.3. Preparation of ZnO NSs/PANI Nano-Porous Film Heterostructure Arrays

PANI nano-porous film was prepared using chemical oxidation of aniline monomer. Firstly, 800 μL of aniline was dissolved in 100 mL of 1 mol/L HCL solution and stirred. Then, 5 mL of 0.1 mol/L ammonium persulfate (APS) solution was dissolved in a mixture of aniline and hydrochloric acid solution with stirring. The resulting mixture was plated in a freezer to keep the reaction temperature around 0–5 °C until the reaction was finally completed. Then, the obtained polyaniline solution was taken out, centrifuged and concentrated, followed by washing with anhydrous ethanol, repeated 2–4 times to prevent residual hydrochloric acid from corroding the ZnO nanosheet arrays. It was then coated on the ZnO nanosheets using a low-speed spin-coating method and dried in air. Finally, a ZnO NSs/PANI nano-porous film heterostructure array was obtained. Then, the two samples were plated face to face and clamped to obtain a double-layer ultraviolet photodetector. Figure 1 shows schematic diagram of experimental assembly and electrochemical testing assembly of ZnO, ZnO/PANI and ZnO/PANI/ZnO heterostructures.

Our work was to prepare a ZnO-based ultraviolet photodetector. Using a three-electrode system and a KCl solution of Zn (NO_3_)_2_ as the electrolyte, ITO/ZnO was prepared by electrodepositing ZnO nanosheets onto ITO conductive glass at −1.1 V and 50 °C for 30 min. Afterwards, PANI was spin-coated onto ITO on the ZnO surface, making ITO/ZnO/PANI. Two identical ITO/ZnO/PANI sheets were prepared, and assembled with the PANI face to face, making ITO/ZnO/PANI/ITO. And a series of morphology, structure and photoelectric performance tests were conducted on it.

### 2.4. Structural Characterizations of Materials

#### 2.4.1. Field Emission Scanning Electron Microscope, FESEM

This project used JSM-7001F field emission scanning electron microscopy from JEOL Company in Tokyo, Japan. Cut the sample into appropriate sizes, place it on a metal sample table, spray gold for 80 s, accelerate the voltage to 15 kV and observe the morphology of the sample after testing.

#### 2.4.2. X-ray Powder Diffraction, XRD

This project used a PW3040/60 X-ray powder diffractometer from Panalytical (Shanghai, China) in the Netherlands, with a Cu Ka target. The X-ray wavelength is 0.15406 nm, the scanning speed is 2°/min, the step width is 0.02°, the test current is 40 mA, the voltage is 40 kV and the scanning angle is 10–80°.

#### 2.4.3. UV–Vis Spectrophotometer

This project uses a UV-2450 ultraviolet visible spectrophotometer from Shimadzu Company in Tokyo, Japan. The light source is a deuterium lamp and a tungsten lamp, with a set ultraviolet wavelength range of 200–800 nm. During the testing process, we first baseline the blank ITO conductive glass as a blank control and then sequentially place the tested ZnO, PANI and ZnO/PANI into the sample cell for testing. During testing, the blank ITO was kept in the blank pool for comparison and, every time the sample was tested, a new baseline was taken to ensure the accuracy of the experiment.

#### 2.4.4. Current Time Curve

This project uses CHI660D electrochemical workstation from Shanghai Chenhua Technology Co., Ltd. (Chenhua, China) to test the current time curve of the device, with an external bias of 0 and a sampling interval of 0.1 s. The ultraviolet source is provided by a portable ultraviolet lamp (ENF-280C, New York, NY, USA). The wavelength is 254 nm and 365 nm. The intensity of long wave (UVA) is 470 μW/cm^2^, medium wave (UVB) intensity 450 μW/cm^2^ and short wave (UVC) intensity 500 μW/cm^2^.

## 3. Results and Discussion

### 3.1. Control of Process Parameters

Figure 2 shows the SEM images of ZnO nanosheet arrays prepared using an anodic oxidation method at different oxidation times, temperatures and potentials. Figure 2a–c show the SEM images of samples under reaction times of 10 min, 30 min and 60 min. After the reaction was carried out for 10 min, a complete array of ZnO nanosheets was prepared, with uniform and stable morphology, indicating a fast reaction rate. After the reaction was carried out for 30 min, the thickness of the obtained ZnO nanosheet array significantly decreased, appearing almost translucent, with a smooth surface and a larger size compared to the sample obtained at 10 min. And it was more conducive to contact with ultraviolet light and improved the utilization of visible light. When the reaction lasted for 60 min, the thickness of the obtained ZnO nanosheet array showed an increasing trend, and its thickness was uneven. The ZnO nanosheet array obtained after 30 min of reaction was more in line with the requirements for its application in optoelectronic devices, with less thickness and a larger specific surface area, which was conducive to contact with the light source and improved the utilization of visible light. Therefore, we set the optimal reaction time as 30 min and, unless otherwise specified in the following text, the default reaction time was 30 min.

Figure 2d–f show the products obtained at 30 °C, 40 ℃ and 50 °C. It can be seen that, at 30 °C, the nanosheet morphology has not yet formed and the ITO substrate surface has an initial sheet-like fold morphology, indicating a trend towards the formation of nanosheet morphology. However, there was not enough temperature to support the reaction. When the temperature rose to 40 °C, a complete array of ZnO nanosheets has been formed but, when compared to the sample at 50 °C, the latter had a larger size and more complete and uniform morphology. Therefore, we chose 50 °C as the ideal reaction temperature and, unless otherwise specified in the following text, it is assumed that the reaction temperature is 50 °C.

Figure 2g–i show the ZnO nanosheet array under oxidation voltages of −0.7 V, −1.1 V and −1.5 V. And it can be seen that, when the oxidation voltage is as low as −0.7 V, a complete array of ZnO nanosheets is not formed on the ITO substrate surface, only gauze-like layers and a small number of massive solids. As the oxidation potential increases, the ZnO nanosheets array begin to form. When the oxidation potential reaches −1.1 V, the complete morphology of the ZnO nanosheet array can be observed on the ITO surface, and its morphology is uniform and stable. When the oxidation potential continues rising to −1.5V, it can be observed that all the ZnO nanosheets on the ITO substrate surface fold and cluster together, forming a flower-like structure. Therefore, we can obtain that −1.1 V is the optimal potential for preparing ZnO nanosheet arrays. Unless otherwise specified in the following text, the default oxidation potential is −1.1 V.

### 3.2. Characterization of ZnO Nanosheets and ZnO/PANI Heterostructures

Figure 3 shows the SEM images of ZnO nanosheets and ZnO/PANI heterostructures on ITO substrates. Figure 3a–c show the top SEM images of ZnO nanosheet arrays, PANI nanowires and the magnification of ZnO/PANI heterostructures, respectively. The size of PANI nano-porous film is approximately 80 nm. Figure 3d shows the cross-sectional SEM image of ZnO/PANI heterostructures. In Figure 3a, it can be seen that the substrate is covered with many vertically arranged nanosheets, with almost uniform diameter and thickness. The prepared ZnO nanosheet array grew well on the surface of ITO conductive glass, and was evenly and tightly distributed on the conductive surface of the ITO substrate. It can be seen that this method eliminated the tedious process of preparing the seed layer and does not affect its growth. The average diameter of ZnO nanosheets is about 4 μm. The thickness is below 100nm. We can also observe that the surface of these two-dimensional hexagonal nanosheets is smooth and the nanosheets cover the entire ITO substrate. Figure 3c,d show complete contact between ZnO nanosheets and PANI nano-porous film. This helps to form p-n heterojunctions between n-type ZnO and p-type PANI, which can improve the sensitivity of the sensor and facilitate the contact between nanocomposites and ultraviolet light, as well as the transfer of photogenerated electrons [41].

The XRD spectra of ZnO nanosheets, PANI nano-porous film and ZnO/PANI heterostructures are shown in Figure 4. ZnO and ZnO/PANI have a hexagonal wurtzite structure. The peak marked with a star is the base peak of the ITO conductive glass. In addition, no diffraction peaks related to PANI were observed in the ZnO/PANI heterostructures, possibly due to the addition of ZnO reducing the X-ray diffraction peaks of PANI, indicating that the introduction of PANI does not affect the crystal structure of ZnO NRs. The sharp diffraction peaks indicate that the synthesized ZnO nanosheets have good crystallinity [42].

The UV–visible absorption spectra of ZnO nanosheets, PANI nano-porous film and ZnO/PANI heterostructures are shown in Figure 5. The ZnO nanosheet array and ZnO/PANI heterostructures exhibit good absorption intensity in the ultraviolet region. ZnO nanosheets exhibit strong absorption peaks in the UV range of 260–380 nm, while PANI nano-porous film and ZnO/PANI heterostructures exhibit strong absorption peaks in the UV range of 310–490 nm. Compared with ZnO nanosheets and PANI nano-porous film, ZnO/PANI heterostructures exhibit stronger absorption peaks in the ultraviolet range of 310–490 nm. This gave us reason to believe that, when ZnO nanosheet arrays were combined with PANI nano-porous film, they had better photosensitivity in the ultraviolet region, which can improve the utilization of light sources and thus improve a series of performances of ultraviolet photodetection devices. Therefore, we chose 254 nm, 312 nm and 365 nm ultraviolet light to test the UV response of the synthesized ZnO nanosheets and ZnO/PANI heterostructures.

### 3.3. Photoelectric Performance Testing of ZnO/PANI Heterostructures

The I-V characteristics of ZnO/PANI heterostructures are shown in Figure 6a, which clearly show a nonlinear and asymmetric I-V curve. The curve shows the voltametric characteristics of ZnO/PANI heterostructures under dark environment and UV irradiation at wavelengths of 254nm and 365nm, respectively. It can be seen that the device exhibited a significant rectification effect under ultraviolet light irradiation, indicating that the ZnO nanosheet arrays formed a p-n heterostructure after recombination with PANI nanowires. At the same time, under the same voltage conditions, the current increased with the increase in ultraviolet wavelength. The rectification behavior was mainly due to the formation of p-n junctions between P-type PANI nano-porous film and N-type ZnO nanosheets. In addition, the current of the device under ultraviolet light was higher than that under a dark environment and the photocurrent of the device under 365 nm ultraviolet light was higher than that under 254 nm ultraviolet light. The results showed that ZnO/PANI heterostructures have UV photosensitivity. In order to further verify the UV photosensitivity of nanocomposites, the I-t curves of nanocomposites under 0 V biased UV irradiation were studied. Figure 6b shows the photocurrent curves of pure ZnO nanosheets and ZnO/PANI heterostructures generated by periodically turning on and off ultraviolet light with a wavelength of 365 nm. The results clearly show that the photocurrents of ZnO NSs and ZnO NSs/PANI nano-porous film increased immediately under UV light and decreased immediately after UV light was switched off. In addition, the addition of PANI increased the photocurrent by two orders of magnitude under the same UV light irradiation. Therefore, we can conclude that, when ZnO nanosheet arrays are combined with PANI nano-porous film, the photoelectric conversion efficiency and visible light utilization rate can be significantly improved, providing a possibility for improving the photoelectric performance of the device.

Figure 6c shows the photocurrent curves of single-layer ZnO/PANI heterostructures and ZnO/PANI/ZnO sandwich-structure devices, with a bias voltage of 0 V and a UV wavelength of 365 nm for irradiation. Compared with single-layer ZnO/PANI heterostructures, devices based on a ZnO/PANI/ZnO sandwich structure have significantly improved photosensitivity and photocurrent. In addition, as shown in Figure 6b,c, the I-V curves of ZnO/PANI show different shapes from the ZnO and ZnO/PANI/ZnO samples. ZnO itself had defects. When light was added, the current immediately increased, indicating that the increased current was generated due to the introduction of light. However, as the current increased, sharp peaks appeared, which may be due to the rapid recombination of electron holes, leading to a sharp decrease in photocurrent and the appearance of sharp peaks. The addition of PANI had to some extent compensated for this defect, leading to a slow increase in current. ZnO/PANI/ZnO also experienced a sudden drop in current, which may be due to the inability of PANI to fully compensate for the large defects in ZnO. But it can also be seen that ZnO/PANI/ZnO sandwich-structure devices had higher photoelectric conversion efficiency, which also provides new ideas for improving the performance of various aspects of ultraviolet photoelectric devices.

In addition, by comparing the switching ratio, responsiveness and rise/fall time of photocurrent devices, as shown in Table 1, it can be found that the switching ratio and responsiveness of ZnO/PANI/ZnO heterostructures are greater than those of ZnO and ZnO/PANI, and the time required for the current peak to rise or decay by 80% is less than that of ZnO and ZnO/PANI. This gives us reason to believe that the photoelectric performance of ZnO/PANI/ZnO heterostructure optoelectronic devices is the best. 

Therefore, our UV photodetectors should be based on the ZnO NSs/PANI nano-porous film/ZnO NSs sandwich structure. The I-t curves of ZnO/PANI/ZnO devices under UV irradiation at wavelengths of 254 nm, 312 nm and 365 nm are shown in Figure 6d. The results indicate that the photocurrent generated by ZnO/PANI/ZnO ultraviolet detectors under different wavelengths of ultraviolet light irradiation gradually increases with the increase in ultraviolet wavelength, and the experimental results are stable and have good repeatability.

### 3.4. Working Principle of ZnO/PANI/ZnO Sandwich-Structure Devices

Figure 7 shows the schematic diagram of the ZnO/PANI p-n junction under ultraviolet light irradiation. When p-type semiconductor PANI and n-type ZnO were compounded in contact with each other, a p-n heterojunction with a vertical structure was formed. In n-type ZnO, most carriers were electrons and a few were holes. In p-type PANI, most carriers were holes and a few were electrons. When n-type ZnO comes into contact with p-type PANI, due to the concentration gradient of charge carriers, electrons will diffuse from the n-region to the p-region, while holes will diffuse from the p-region to the n-region, which will make one side of the n-region positively charged and the other side of the p-region negatively charged, so that the direction of the internal electric field is from ZnO to PANI. This region is called the space charge region [43,44,45,46]. When UV radiation is irradiated to the device, many electron–hole pairs can be generated, and the separated electrons and holes quickly drift to the two electrodes under the action of an external electric field. As a result, the excited electrons are unpaired, so the current through the junction and around the circuit increases.

## 4. Conclusions

In summary, the research in this paper lay in the successful fabrication of a sensitive ultraviolet (UV) photodetector based on the p-n junction structure consisting of ZnO nanosheets and PANI nano-porous film. The device was easy and convenient to make. The preparation of ZnO adopted a simple and feasible anodic oxidation method, which avoided the cumbersome process of preparing seed layers in traditional methods. The obtained samples had uniform size, larger specific surface area than other morphology nano ZnO and can better make contact with the light source, resulting in more contact sites. The preparation of PANI was carried out using a low-temperature chemical oxidation method. After sampling, it was repeatedly washed and centrifuged with anhydrous ethanol 2–3 times, and then composited with a ZnO nanosheet array. After a series of photoelectric performance tests, the results showed that, under the same duration of ultraviolet light irradiation, the UV photocurrent of ZnO/PANI devices was 100 times higher than that of pure ZnO devices. At a wavelength of 365 nm, the device exhibited high photocurrent responsiveness and photoconductivity. The good device performance was attributed to the large specific surface area of ZnO nanosheets and the p-n junction formed between p-type PANI nano-porous film and n-type ZnO nanosheets. Moreover, due to the p-n heterostructure formed by the combination of ZnO nanosheet arrays and PANI nano-porous film, its photoelectric performance was superior to that of pure ZnO nanosheet arrays. At the same time, the photoelectric performance of ZnO/PANI/ZnO sandwich-structure devices was superior to that of ZnO/PANI double-layer structure composite materials. This provides a solid theoretical basis for the application of ZnO nanosheets in ultraviolet detection, and these findings have certain guiding significance for the development of ultraviolet photodetectors, and provide new ideas for optimizing the performance and structure of future UV optoelectronic devices.

## Figures and Tables

**Figure 1 polymers-15-04399-f001:**
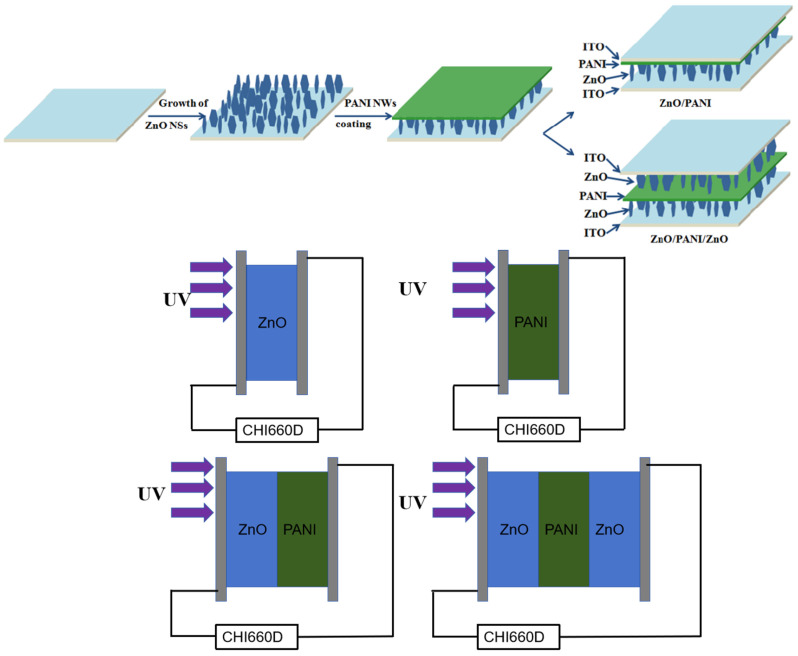
Schematic diagram of experimental assembly and electrochemical testing assembly of ZnO, ZnO/PANI and ZnO/PANI/ZnO heterostructures.

**Figure 2 polymers-15-04399-f002:**
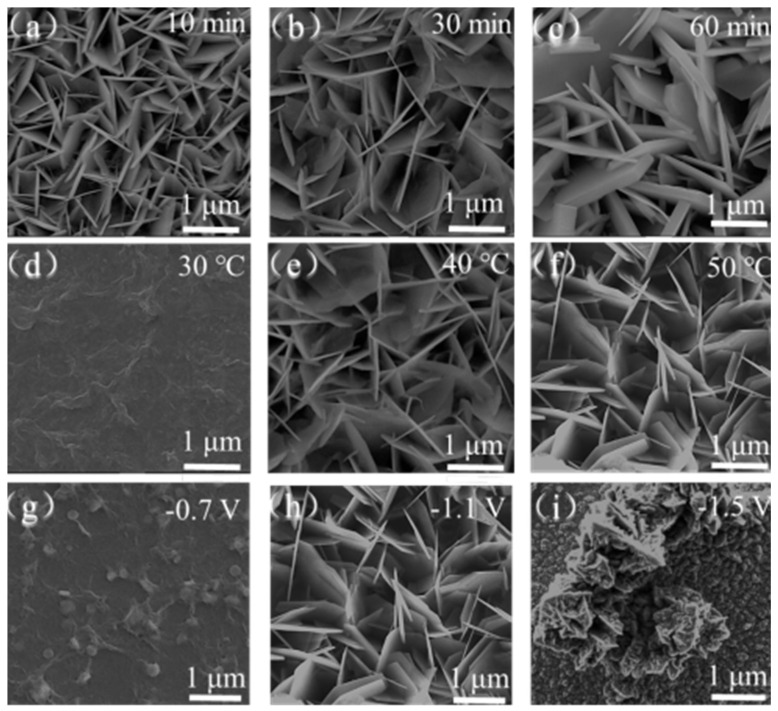
SEM images of ZnO nanosheet array preparation under different oxidation times (**a**–**c**), temperatures (**d**–**f**) and voltages (**g**–**i**).

**Figure 3 polymers-15-04399-f003:**
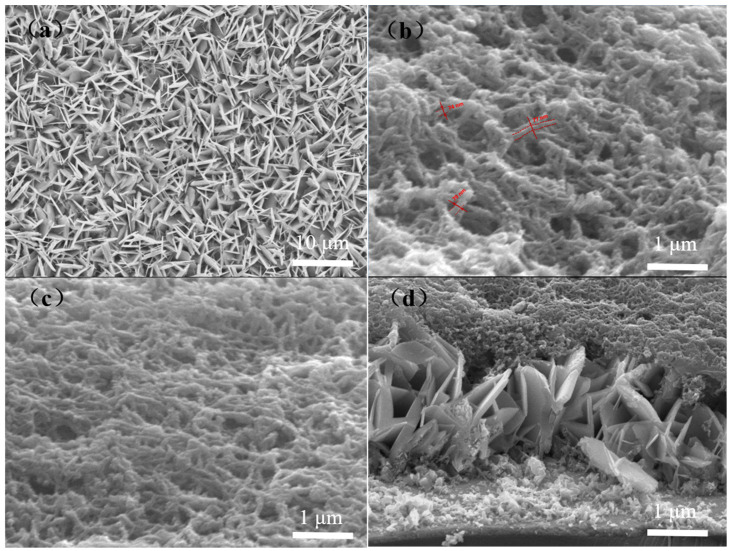
SEM images of ZnO nanosheet array (**a**), PANI nano-porous film (**b**) and ZnO/PANI heterostructures (**c**,**d**).

**Figure 4 polymers-15-04399-f004:**
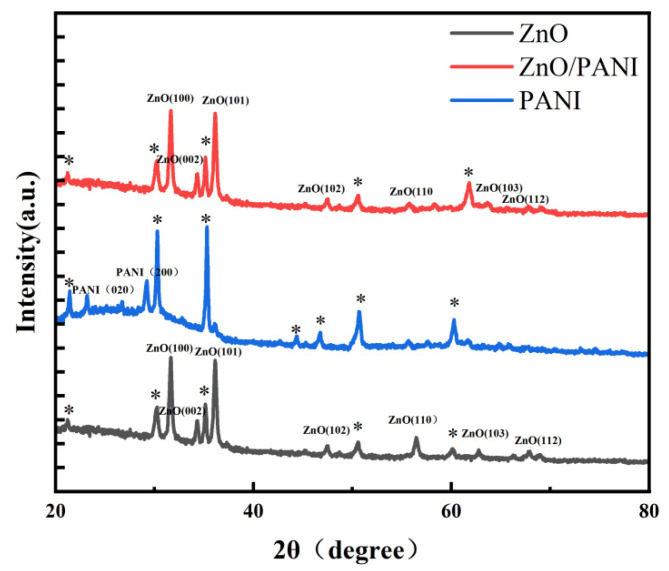
XRD pattern of ZnO nanosheets, PANI nano-porous film and ZnO/PANI heterostructures (* base peak of ITO conductive glass).

**Figure 5 polymers-15-04399-f005:**
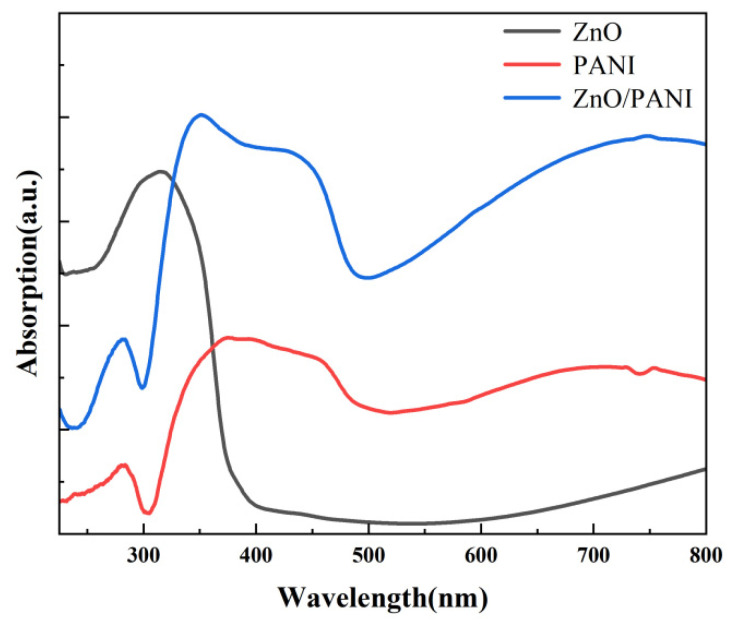
UV–vis absorption spectra of ZnO nanosheets, PANI nano-porous film and ZnO/PANI heterostructures.

**Figure 6 polymers-15-04399-f006:**
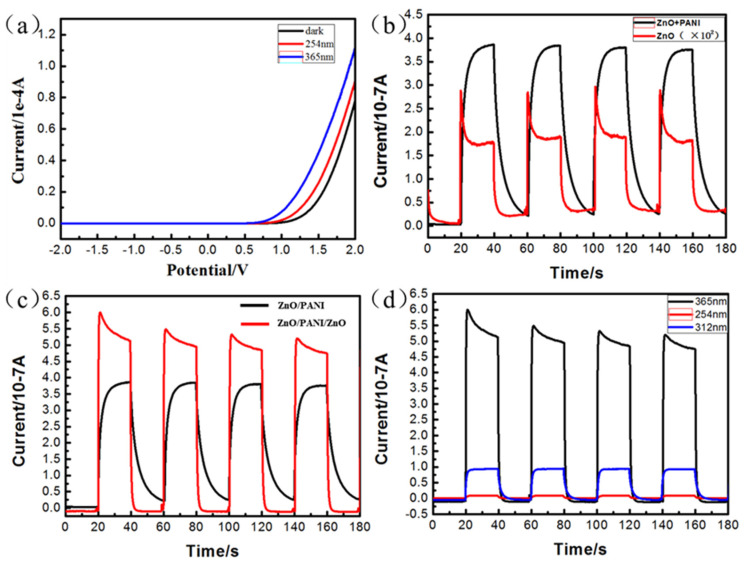
Voltammetric curves (**a**) of ZnO/PANI heterostructures under different wavelengths of ultraviolet light; I-t curves of ZnO nanosheet arrays and ZnO/PANI heterostructures (**b**); I-t curves of ZnO/PANI and ZnO/PANI/ZnO heterostructures (**c**); I-t curves of ZnO/PANI/ZnO heterostructures under different wavelength ultraviolet irradiation (**d**).

**Figure 7 polymers-15-04399-f007:**
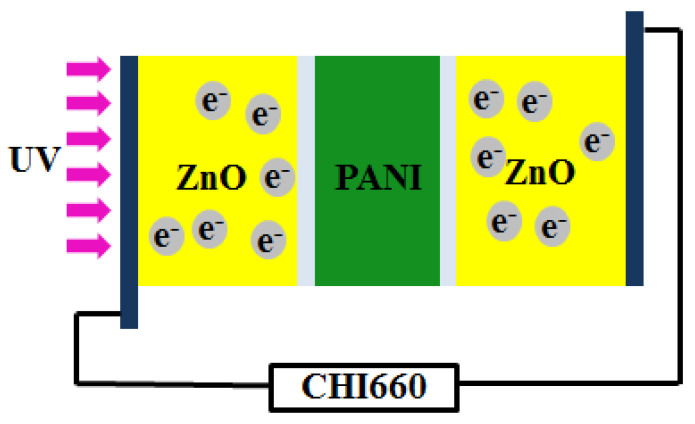
Mechanism diagram of ZnO/PANI/ZnO sandwich-structure device.

**Table 1 polymers-15-04399-t001:** On–off ratio, responsiveness and rise/fall schedule.

Sample	On–Off Ratio	Responsiveness S(A/W)	Rise Time (80%)/s	Fall Time (80%)/s
ZnO	9.27	0.000465	20.4	28.5
ZnO/PANI	33.46	0.00622	13.0	22.9
ZnO/PANI/ZnO	60.34	0.00968	5.4	20.3

## Data Availability

The data presented in this study are available on request from the corresponding author.

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
