# Peer review of "Preparation of ZnO Nanosheet Array and Research on ZnO/PANI/ZnO Ultraviolet Photodetector"

_polymers, 2023, doi:10.3390/polym15224399_

Round 1

Reviewer 1 Report

Comments and Suggestions for Authors

The research article entitled “Preparation of ZnO nanosheet array and research of ZnO/PANI/ZnO ultraviolet photodetector” needs to address following queries.

1. The introduction part requires Improvement and more focused

2. What indicate Star symbol (*) point in the XRD pattern of ZnO in Fig. 4 page 8.

3. Authors also need to incorporate XRD pattern of PANI and ZnO-PANI composite.

4. UV-Visible absorption of PANI is not given. In most of the cases, after incorporating the ZnO into PANI matrix. Absorbance decreases, but in your case, it increases. An explanation of this is needed.

5. Which procedure was carried out for taking UV-Vis absorption spectra, mention this in section 2.4.3.

6. If the intensity of the light source is the same, then photocurrent under a lower wavelength must be higher, but in your case, as in Fig. 6(a) it is not. Why? This needs a detailed explanation.

7. Authors mention this statement “under the same voltage conditions, the current in-289 creased with the increase of ultraviolet wavelength.” Is it possible?

8. In case of a photodetector there are many figures of merits, based on which we can evaluate the photoperformance. Authors need to examine those parameters such as on-off ratio, responsivity, detectivity, noise equivalent power, linear dynamic range, external quantum efficiency etc. Only based on photocurrent you cannot justify your results.

9. To better understand the carrier generation and recombination, rise and decay time constants can also be calculated.

10. Authors need to mention the type of light source and intensity of light in section 2.4.

Manuscript may be accepted after thorough major revision.

Comments on the Quality of English Language

Need to be improved.

Author Response

Reviewer #1:

The   research   article    entitled   “Preparation    of   ZnO   nanosheet    array   and    research   of ZnO/PANI/ZnO ultraviolet photodetector” needs to address following queries.

  1. The introduction part requires Improvement and more focused

Answer:Thank you for your comments. Based on your constructive suggestions, we have supplemented the introduction with a detailed description of the relevant research work in this field, as well as further highlighting the differences between this work and other studies, highlighting the advantages of this work. We have revised the expressions and sincerely hope that the answers would meet your requirements. Thanks again for your professional comments.

Original:

Advances in functional nanomaterials, nanofabrication technology, and device architecture have stimulated the development of semiconductor photodetectors with greater miniaturization, lower power consumption, more functionality, and higher response speed and accuracy. The development of wide band gap semiconductor technology has aroused extensive attention and interest in ultraviolet optoelectronic devices [1]. Especially in recent years, ultraviolet photoelectric detectors have been widely used in communication detection [2], aerospace [3], medicine [4], biology [5]and other fields. As is well known, traditional ultraviolet photodetectors typically require the application of an external bias voltage to separate photo generated charge carriers and enhance their photoresponse characteristics. Broadband gap oxide semiconductor ultraviolet photodetectors are based on p-n junctions, which convert ultraviolet radiation into electrical signals through photovoltaic effects. Compared with traditional bulk materials or thin films, UV photodetectors built on semiconductor materials, especially one-dimensional nanostructures, such as nanoribbons, nanotubes and nanowires, which usually have highly competitive performance advantages due to their volumetric light absorption. And these devices have lower power consumption, more functions, and higher responsiveness and accuracy [6-10].

 Nanostructured oxide semiconductor materials exhibit superior and unique physical properties due to their large surface area to volume ratio and quantum confinement effect [11-13]. Therefore, one-dimensional photodetectors made of ZnO wide bandgap nano oxide semiconductor materials have become a widely studied topic due to their strong device performance [14-18]. ZnO is a wide bandgap semiconductor material with a direct bandgap of 3.37 eV [19-20] and an exciton binding energy of up to 60 meV [21]. It has good thermal and chemical stability, and is low-cost and easy to obtain. It has been identified as one of the most effective antireflective semiconductor materials in optical detection applications [22,23]and is widely used in various fields such as ultraviolet optoelectronic devices [24], solar cells [25], and light-emitting diodes [26]. However, ZnO is susceptible to corrosion by light, resulting in low visible light utilization efficiency, high recombination rate of photogenerated electron hole pairs, and low quantum efficiency and difficulty in obtaining p-ZnO with stable performance due to the inherent self-compensation effect of its donor. This makes it almost impossible to achieve high-efficiency ultraviolet photodetectors based on ZnO homojunction. [27-29]. Compared with bulk structures, the nanostructured form of ZnO provides excellent optical and electrical properties, and the composite modification of nanostructured ZnO with semiconductors can reduce the recombination rate of photogenerated electron hole pairs, which  improves the quantum efficiency[30]. In conductive polymers, the high conductivity of the polymer can promote rapid charge transfer, thereby improving the overall performance of the device. PANI is used as a conductive material and ZnO nanosheets for composite modification due to its unique redox properties that can be modified to meet specific requirements by changing synthesis conditions or doping PANI with various components, as well as significant electrochemical performance, environmental stability, excellent conductivity, high theoretical capacitance, and low cost [31-33].

However, the preparation process of ZnO nanosheets has been complicated, and the experimental reproducibility and sample uniformity are not high for a long time, which have become obstacles to the application of ZnO nanosheets in optoelectronic components [34]. In recent years, many different technologies for synthesizing ZnO nanostructures have been reported both domestically and internationally, such as electrodeposition, chemical oxidation, and chemical bath deposition. Electrochemical deposition, known as electrodeposition or electroplating, is a popular chemical method for growing crystalline seed layers. This method can improve deposition quality and enhance adhesion, which is greatly helpful for synthesizing different forms of ZnO nanostructures. Due to the fact that electrochemical deposition only allows the growth of nanostructures when voltage is applied, and only at the working electrode, this method can effectively reduce the loss of chemical materials, save resources, and eliminate waste, which is in line with the national sustainable development strategy. Other advantages of electrochemical deposition method include easy stoichiometric control by adjusting the deposition voltage, the ability to control the deposition rate by adjusting the applied current density, ease of doping semiconductors, and the possibility of designing or manipulating the bandgap of nanomaterials[35].In this work, a uniform ZnO nanoparticle array was directly grown on the surface of ITO conductive glass by means of electrodeposition and chemical oxidation [36], eliminating the complicated process of preparing the traditional ZnO seed layer. A unique "sandwich" structure UV photoelectric detection device based on ZnO nanosheets array was designed and prepared, and a series of photoelectric performance tests were carried out. The results showed that the UV current of ZnO/PANI devices was 100 times higher than that of pure ZnO devices under the same UV irradiation time. At 365 nm wavelength, the device had excellent photocurrent responsiveness and photoconductivity.

Revised:

Ultraviolet (UV) radiation, with a wavelength of 10-400 nm, is one of the strongest radiation in nature and has a profound impact on the survival and evolution of life on Earth. Therefore, the increasing demand for ultraviolet radiation monitoring in many fields has promoted the research of various ultraviolet photodetectors [1].Especially in recent years, ultraviolet photoelectric detectors have been widely used in communication detection [2], aerospace [3], medicine [4], biology [5]and other fields.Traditional ultraviolet photodetectors typically require the application of an external bias voltage to separate photo generated charge carriers and enhance their photoresponse characteristics. Broadband gap oxide semiconductor ultraviolet photodetectors are based on p-n junctions, which convert ultraviolet radiation into electrical signals through photovoltaic effects. Compared with traditional bulk materials or thin films, UV photodetectors built on semiconductor materials, especially one-dimensional nanostructures, such as nanoribbons, nanotubes and nanowires, which usually have highly competitive performance advantages due to their volumetric light absorption. And these devices have lower power consumption, more functions, and higher responsiveness and accuracy [6-10].

Nanostructured oxide semiconductor materials exhibit superior and unique physical properties due to their large surface area to volume ratio and quantum confinement effect [11-13]. Therefore, one-dimensional photodetectors made of ZnO wide bandgap nano oxide semiconductor materials have become a widely studied topic due to their strong device performance [14-18]. ZnO is a wide bandgap semiconductor material with a direct bandgap of 3.37 eV [19-20] and an exciton binding energy of up to 60 meV [21]. It has good thermal and chemical stability, and is low-cost and easy to obtain. It has been identified as one of the most effective antireflective semiconductor materials in optical detection applications [22,23]and is widely used in various fields such as ultraviolet optoelectronic devices [24], solar cells [25], and light-emitting diodes [26]. However, ZnO is susceptible to corrosion by light, resulting in low visible light utilization efficiency, high recombination rate of photogenerated electron hole pairs, and low quantum efficiency and difficulty in obtaining p-ZnO with stable performance due to the inherent self-compensation effect of its donor. This makes it almost impossible to achieve high-efficiency ultraviolet photodetectors based on ZnO homojunction. [27-29]. Compared with bulk structures, the nanostructured form of ZnO provides excellent optical and electrical properties, and the composite modification of nanostructured ZnO with semiconductors can reduce the recombination rate of photogenerated electron hole pairs, which  improves the quantum efficiency[30].

Based on the characteristics and advantages of ZnO, it has aroused widespread interest among researchers.Huang[31] prepared ultraviolet photodetectors based on ZnO nanorods and Al nanoparticles. Research had found that the photoelectric performance of ZnO based ultraviolet photodetectors is significantly improved after the addition of Al nanoparticles. The mechanism is that LSPR occurs when Al nanoparticles are irradiated on ZnO nanorods, resulting in enhanced absorption. Grigoryev, LV et al.[32] reported the research results on the structure, optics, photoluminescence, and optoelectronic properties of ZnO-LiNbO3 thin film structure. The X-ray structural analysis results of ZnO films synthesized on single crystal lithium niobate substrate and quartz glass substrate were introduced. The transmittance, reflectance, absorption, photoluminescence, and photocurrent spectra of ZnO-LiNbO3 thin film and ZnO-SiO2 structure in the ultraviolet and visible spectral regions were given.Yin et al.[33] synthesized Cu and Ag co doped ZnO nanorod arrays on p-GaN/Al2O3 substrates using low-temperature hydrothermal method, and studied the effects of co doping on morphology, microstructure, and electrical/optical properties. Compared with other samples, the average diameter of co doped ZnO NRs is larger, and the unit area (density) of NRs is less. Ag+Cu co doping can also reduce the band gap of ZnO NRs. And Ag+Cu co doping can greatly improve the electrical performance of heterojunctions, demonstrating the potential application of Ag+Cu co doping ZnO NRs in ultraviolet light-emitting diodes.

In conductive polymers, the high conductivity of the polymer can promote rapid charge transfer, thereby improving the overall performance of the device. PANI is used as a conductive material and ZnO nanosheets for composite modification due to its unique redox properties that can be modified to meet specific requirements by changing synthesis conditions or doping PANI with various components, as well as significant electrochemical performance, environmental stability, excellent conductivity, high theoretical capacitance, and low cost [34-36].Muhammad Naveed ur Rehman et al.[37]  prepared raw ZnO, Y2O3, binary PANI-Y2O3, PANI-ZnO, Y2O3-ZnO, and novel ternary PANI-Y2O3-ZnO nanocomposites using co precipitation and ultrasound techniques in this study. It has been proven that the highest specific capacitance value of the new ternary nanocomposite material is due to the fast charge transfer rate and enhanced surface dependent electrochemical performance of PANI.

  1. What indicate Star symbol (*) point in the XRD pattern of ZnO in Fig. 4 page

Answer:Thank you very much for your guidance and I’m pleased to receive your modification suggestions.The peak marked with stars is the base peak of ITO conductive glass. I will explain it in the text. I apologize for forgetting to explain it in the manuscript, which caused you reading difficulties.We sincerely hope that the answers would meet your requirements. Thanks again for your professional comments.

Original:

The XRD pattern of ZnO nanosheets was shown in Figure 4. All the diffraction peaks can point to a hexagonal fibrillar zincite structure. Except for the diffraction peaks of ITO substrate, no characteristic peaks of other impurities were detected, which indicated that only single-phase ZnO samples have been formed. Sharp diffraction peaks indicated good crystallinity of the synthesized ZnO nanosheets [38].

Revised:

The XRD pattern of ZnO nanosheets was shown in Figure 4. All the diffraction peaks can point to a hexagonal fibrillar zincite structure. The peak marked with stars is the base peak of ITO conductive glass.Except for the diffraction peaks of ITO substrate, no characteristic peaks of other impurities were detected, which indicated that only single-phase ZnO samples have been formed. Sharp diffraction peaks indicated good crystallinity of the synthesized ZnO nanosheets [38].

  1. Authors also need to incorporate XRD pattern of PANI and ZnO-PANI composite.

Answer:Thank you for your professional comments. The XRD spectra of PANI and ZnO/PANI have been provided in the manuscript.All diffraction peaks can point towards the hexagonal wurtzite structure. The peak marked with a star is the base peak of ITO conductive glass. Except for the diffraction peaks of ITO substrate, no characteristic peaks of other impurities were detected, indicating that only single-phase ZnO samples were formed.. No diffraction peaks related to PANI were observed in ZnO/PANI nanocomposites, indicating that the introduction of PANI and ultrasonic treatment did not affect the crystal structure of ZnO NRs. The sharp diffraction peaks indicate that the synthesized ZnO nanosheets have good crystallinity

Original:

The XRD pattern of ZnO nanosheets was shown in Figure 4. All the diffraction peaks can point to a hexagonal fibrillar zincite structure. The peak marked with stars is the base peak of ITO conductive glass. Except for the diffraction peaks of ITO substrate, no characteristic peaks of other impurities were detected, which indicated that only single-phase ZnO samples have been formed. Sharp diffraction peaks indicated good crystallinity of the synthesized ZnO nanosheets [42].

Figure 4. XRD pattern of ZnO nanosheets

Revised:

The XRD spectra of ZnO nanosheets, PANI nanowires, and ZnO/PANI heterostructures are shown in Figure 4. ZnO and ZnO/PANI have a hexagonal wurtzite structure. The peak marked with a star is the base peak of ITO conductive glass. In addition, no diffraction peaks related to PANI were observed in the ZnO/PANI nanocomposites, possibly due to the addition of ZnO reducing the X-ray diffraction peaks of PANI, indicating that the introduction of PANI does not affect the crystal structure of ZnO NRs. The sharp diffraction peaks indicate that the synthesized ZnO nanosheets have good crystallinity [42].

Figure 4. XRD pattern of ZnO nanosheets,PANI nanowires,ZnO/PANI heterostructure .

  1. UV-Visible absorption of PANI is not given. In most of the cases, after incorporating the ZnO into PANI matrix. Absorbance decreases, but in your case, it increases. An explanation of this is

Answer:Thank you for your comments.We added the UV visible absorption of PANI, and for the sake of experimental rigor, we re prepared samples of ZnO and ZnO/PANI, and measured their UV visible absorption together. The results showed that after doping PANI into ZnO substrate, the absorbance increased. The reason may be that the absorption wavelength of ultraviolet is between 200-380nm, and when light is absorbed within this wavelength range, the valence electrons in polyaniline molecules will undergo transitions. In addition, the energy of ultraviolet light excites the electrons on the surface of ZnO, making them high-energy electrons, i.e. electron hole pairs, which combine with PANI to enhance its ultraviolet absorption ability.We hope that our explanation can be accepted by you and thank you again for your very professional and meticulous comments.

Original:

The UV visible absorption spectra of ZnO nanosheets and ZnO/PANI heterostructure were shown in Figure 5. ZnO nanosheet arrays and ZnO/PANI heterostructure had good absorption strength in the ultraviolet region. ZnO nanosheets and ZnO/PANI heterostructure exhibited strong absorption peaks in the UV band of 220-400 nm, where the stronger absorption peaks in the UV band of 300-400 nm of ZnO/PANI heterostructure were stronger than that of ZnO nanosheets. It gived us reason to believe that when ZnO nanosheet arrays are combined with PANI nanowires, they had better photosensitivity in the ultraviolet region, which could improve the utilization of light sources and thus improve a series of performance of ultraviolet photodetection devices. Therefore, we chose 254 nm, 312 nm, and 365 nm ultraviolet light to test the UV response of the synthesized ZnO nanosheets and ZnO/PANI heterostructure.

Figure 5. UV-VIS absorption spectra of ZnO nanosheets and ZnO/PANI heterostructure .

Revised:

The UV visible absorption spectra of ZnO nanosheets, PANI nanowires, and ZnO/PANI heterostructures are shown in Figure 5. The ZnO nanosheet array and ZnO/PANI heterostructure exhibit good absorption intensity in the ultraviolet region. ZnO nanosheets exhibit strong absorption peaks in the UV range of 260-380nm, while PANI nanowires and ZnO/PANI heterostructures exhibit strong absorption peaks in the UV range of 310-490nm. Compared with ZnO nanosheets and PANI nanowires, ZnO/PANI heterostructures exhibit stronger absorption peaks in the ultraviolet range of 310-490nm. This gives us reason to believe that when ZnO nanosheet arrays are combined with PANI nanowires, they have better photosensitivity in the ultraviolet region, which can improve the utilization of light sources and thus improve a series of performance of ultraviolet photodetection devices. Therefore, we chose 254nm, 312nm, and 365nm ultraviolet light to test the UV response of the synthesized ZnO nanosheets and ZnO/PANI heterostructures.

Figure 5. UV-VIS absorption spectra of ZnO nanosheets,PANI nanowires,ZnO/PANI  heterostructure .

  1. Which procedure was carried out for taking UV-Vis absorption spectra, mention this in section 4.3.

Answer:Thank you very much for your guidance. The set UV wavelength range is 200~800 nm.The light source is a deuterium lamp and a tungsten lamp .

Original:

The UV visible absorption spectrum of molecules is generated by the electronic energy level transition of certain functional groups in the molecule after absorbing UV visible radiation. Due to the different molecular, atomic, and molecular spatial structures of various substances, their absorption of light energy is also different. Therefore, each substance has its own unique and fixed absorption spectrum curve, and the content of the substance can be judged or measured based on the absorbance at certain characteristic wavelengths on the absorption spectrum.

This project used a UV-2450 ultraviolet visible spectrophotometer from Shimadzu Company in Japan.

Revised:

This project used a UV-2450 ultraviolet visible spectrophotometer from Shimadzu Company in Japan.The light source is a deuterium lamp and a tungsten lamp, with a set ultraviolet wavelength range of 200-800 nm. During the testing process, we first baseline the blank ITO conductive glass as a blank control, and then sequentially place the tested ZnO, PANI, ZnO/PANI into the sample cell for testing. During testing, the blank ITO was kept in the blank pool for comparison, and every time the sample was tested, a new baseline was taken to ensure the accuracy of the experiment.

  1. If the intensity of the light source is the same, then photocurrent under a lower wavelength must be higher, but in your case, as in Fig. 6(a) it is not. Why? This needs a detailed

 Answer:Thank you for your  professional comments.When ultraviolet light passes through ITO conductive glass, the penetration force varies due to different wavelengths. The larger the wavelength, the smaller the penetration force and energy, and the greater the energy consumed when passing through ITO, resulting in an increase in energy loss. At this point, the smaller the energy of a single photon, the more photons, the more photoelectrons produced, and the higher the saturation current.We hope that our explanation can be accepted by you and thank you again for your very professional and meticulous comments.

  1. Authors mention this statement “under the same voltage conditions, the current in-289 creased with the increase of ultraviolet wavelength.” Is it possible?

Answer:Thank you for your comments.One reason may be that long wavelength ultraviolet radiation has stronger penetration than short wavelength ultraviolet radiation. Another reason may be that the prepared ZnO and ZnO/PANI exhibit stronger absorption at longer wavelengths of UV, as shown in Figure 6 (b, c). The results show that the photoelectric current of ZnO/PANI devices is about 151 times that of ZnO devices, and the photoelectric current of ZnO/PANI/ZnO devices is about 1.55 times that of ZnO/PANI devices.We hope that our explanation can be accepted by you and thank you again for your very professional and meticulous comments.

  1. In case of a photodetector there are many figures of merits, based on which we can evaluate the Authors  need  to  examine  those  parameters  such  as  on-off  ratio, responsivity,  detectivity,  noise   equivalent  power,   linear  dynamic  range,   external  quantum efficiency etc. Only based on photocurrent you cannot justify your results.

Answer:Thank you very much for your guidance. The switching ratios of ZnO, ZnO/PANI, and ZnO/PANI/ZnO are 9.27, 33.46, and 60.34, respectively.The responsiveness of ZnO, ZnO/PANI, and ZnO/PANI/ZnO are  0.000465 A/W, 0.00622 A/W, and 0.00968 A/W, respectively.The switching ratio and responsiveness of ZnO/PANI/ZnO are the highest, so we have reason to believe that the photoelectric performance of ZnO/PANI/ZnO heterostructured optoelectronic devices is the best.

  1. To better understand the carrier generation and recombination, rise and decay time constants can also be

Answer:Thank you very much for your guidance. The photocurrent of ZnO/PANI devices is 134 times that of ZnO devices, and the photocurrent of ZnO/PANI/ZnO devices is 1.6 times that of ZnO/PANI. The photo response time of ZnO increased to 80% of the peak within 20.4 seconds, the photo response time of ZnO/PANI devices rapidly increased to 80% of the peak within 13 seconds, and the photo response time of ZnO/PANI/ZnO devices rapidly increased to 80% of the peak within 5.4 seconds. The photo response time of ZnO increased to 80% of the peak within 20.4 seconds, the photo response time of ZnO/PANI devices rapidly increased to 80% of the peak within 13 seconds, and the photo response time of ZnO/PANI/ZnO devices rapidly increased to 80% of the peak within 5.4 seconds. That is to say, photodetectors based on ZnO/PANI/ZnO nanocomposites have enhanced light response and faster light response time compared to photodetectors based on pure ZnO and ZnO/PANI.We hope that our explanation can be accepted by you and thank you again for your very professional and meticulous comments.

Original:

Figure 6(c) showed the photocurrent curves of single-layer ZnO NSs/PANI NWs nanocomposites and ZnO NSs/PANI NWs/ZnO NSs sandwich structure devices, with a bias voltage of 0 V and a UV wavelength of 365 nm for irradiation. Compared with single-layer ZnO NSs/PANI NWs nanocomposites, devices based on ZnO NSs/PANI NWs/ZnO NSs sandwich structure have significantly improved photosensitivity and photocurrent. In addition, compared to traditional double-layer structured nanocomposites, ZnO/PANI/ZnO sandwich structures exhibit greater photocurrent and high experimental repeatability. It can be seen that ZnO/PANI/ZnO sandwich structure devices have higher photoelectric conversion efficiency, which also provides new ideas for improving the performance of various aspects of ultraviolet photoelectric devices.Therefore, our UV photodetectors should be based on the ZnO NSs/PANI NWs/ZnO NSs sandwich structure. The I-t curves of ZnO NSs/PANI NWs/ZnO NSs devices under UV irradiation at wavelengths of 254 nm, 312 nm, and 365 nm are shown in Figure 6 (d). The results indicated that the photocurrent generated by ZnO/PANI/ZnO ultraviolet detectors under different wavelengths of ultraviolet light irradiation gradually increases with the increase of ultraviolet wavelength, and the experimental results were stable and have good repeatability.

Revised:

Figure 6(c) showed the photocurrent curves of single-layer ZnO NSs/PANI NWs nanocomposites and ZnO NSs/PANI NWs/ZnO NSs sandwich structure devices, with a bias voltage of 0 V and a UV wavelength of 365 nm for irradiation. Compared with single-layer ZnO NSs/PANI NWs nanocomposites, devices based on ZnO NSs/PANI NWs/ZnO NSs sandwich structure have significantly improved photosensitivity and photocurrent.In addition, as shown in Figure 6. (b) And (c), the I-V curves of ZnO/PANI show different shapes from ZnO and ZnO/PANI/ZnO samples. ZnO itself has defects. When light is added, the current immediately increases, indicating that the increased current is generated due to the introduction of light. However, as the current increases, sharp peaks appear, which may be due to the rapid recombination of electron holes, leading to a sharp decrease in photocurrent and the appearance of sharp peaks. The addition of PANI has to some extent compensated for this defect, leading to a slow increase in current. ZnO/PANI/ZnO also experienced a sudden drop in current, which may be due to PANI's inability to fully compensate for the large defects in ZnO. But it also can be seen that ZnO/PANI/ZnO sandwich structure devices have higher photoelectric conversion efficiency, which also provides new ideas for improving the performance of various aspects of ultraviolet photoelectric devices.

In addition, by comparing the switching ratio, responsivity, and rise/fall time of photocurrent devices, as show in Table 1,it can be found that the switching ratio and responsivity of ZnO/PANI/ZnO heterostructures are greater than those of ZnO and ZnO/PANI, and the time required for the current peak to rise or decay by 80% is less than that of ZnO and ZnO/PANI. This gives us reason to believe that the photoelectric performance of ZnO/PANI/ZnO heterostructure optoelectronic devices is the best.

Therefore, our UV photodetectors should be based on the ZnO NSs/PANI NWs/ZnO NSs sandwich structure. The I-t curves of ZnO NSs/PANI NWs/ZnO NSs devices under UV irradiation at wavelengths of 254 nm, 312 nm, and 365 nm are shown in Figure 6 (d). The results indicated that the photocurrent generated by ZnO/PANI/ZnO ultraviolet detectors under different wavelengths of ultraviolet light irradiation gradually increases with the increase of ultraviolet wavelength, and the experimental results were stable and have good repeatability.

Table 1:On-off ratio, Responsiveness, Rise/Fall schedule

Sample

On-off ratio

Responsiveness

(A/W)

Rise time

(80%)/s

Fall time

(80%)/s

ZnO

9.27

0.000465

20.4

28.5

ZnO/PANI

33.46

0.00622

13.0

22.9

ZnO/PANI/ZnO

60.34

0.00968

5.4

20.3

  1. Authors need to mention the type of light source and intensity of light in section 2.4.

Answer:Thank you very much for your comment. The ultraviolet source is provided by a portable ultraviolet lamp (ENF-280C, USA). The wavelength is 365nm. We hope that our explanation can be accepted by you and thank you again for your very professional and detailed comments.

Original:

The current time curve is a method of recording the variation of electrode current with time under a large step potential applied to the working electrode on a stationary electrode and in an unstirred solution (The potential that jumps from a potential without Faraday reaction to the surface active component of the electrode effectively approaches zero).

This project used CHI660D electrochemical workstation from Shanghai Chenhua Technology Co., Ltd. to test the current time curve of the device, with an external bias of 0 and a sampling interval of 0.1 seconds.

Revised:

This project used CHI660D electrochemical workstation from Shanghai Chenhua Technology Co., Ltd. to test the current time curve of the device, with an external bias of 0 and a sampling interval of 0.1 seconds. The ultraviolet source is provided by a portable ultraviolet lamp (ENF-280C, USA). The wavelength is 254 nm and 365nm. The intensity of long wave (UVA) is 470 μW/cm2, medium wave (UVB) intensity of 450 μW/cm2, short wave (UVC) intensity of 500 μW/cm2.

Reviewer 2 Report

Comments and Suggestions for Authors

In this article, ZnO nanosheets and ZnO/PANI/ZnO heterostructures were fabricated for UV photodetector application. Surface SEM images of the fabricated ZnO nanosheets and heterostructures were taken, UV-VIS spectrum analysis was performed and photodetector properties were investigated. Although the title of the paper is intriguing and it is interesting to examine an application of heterostructures, the paper is sloppy and contains many errors. So, in my opinion, the manuscript in present form is not valuable and acceptable for publication in the journal. Some suggestions are given below;

·       Please clarify your devices ITO/PANI/ITO, ITO/ZnO/ITO, ITO/ZnO/PANI/ITO, ITO/ZnO/PANI/ZnO/ITO, …etc.

·       A bare PANI should be prepared for photodetector application. A comparison between bare ZnO, bare PANI, ZnO/PANI and ZnO/PANI/ZnO will be valuable.

·       The authors coated PANI on ZnO nanosheet by using spin coating method as they mentioned in experimental section. But they wrote in Results and Discussion section the PANI in the form of nanowire. Is it right? Please give details of nanowire dimensions with SEM or FESEM images.

·       The authors should make it clear in the Experimental method section which fabrication conditions of the ZnO nanosheet structures were investigated. Figure 2 shows the SEM images for ZnO nanosheets prepared under different conditions.

·       In this study, it should be made clear whether you have created a ZnO/PANI nanocomposite or a heterostructure. In the text, the term composite is used in some places and heterostructure in others.

·       The authors mentioned about Linear sweep voltammetry and AC impedance in the Experimental section. But they did not give any results

·       In my opinion it is not necessary to define analyze or characterization methods in Experimental section such as “The X-ray powder diffractometer is mainly used to study and identify the composition and atomic level structure of substances and materials. It analyzes the composition, 147 particle size, crystallinity, etc. of nanoparticles by comparing them with the diffraction  patterns in the powder diffraction file (PDF) database. From the powder diffraction spectrum, physical quantities such as diffraction peak position, intensity, and peak shape can be directly obtained.

·       The authors have repeatedly used the phrase “This project used” in Experimental section. Why?

Author Response

Reviewer #2:

In this article, ZnO nanosheets and ZnO/PANI/ZnO heterostructures were fabricated for UV photodetector application. Surface SEM images of the fabricated ZnO nanosheets and heterostructures were taken, UV-VIS spectrum analysis was performed and photodetector properties were investigated. Although the title of the paper is intriguing and it is interesting to examine an application of heterostructures, the paper is sloppy and contains many errors. So, in my opinion, the manuscript in present form is not valuable and acceptable for publication in the journal. Some suggestions are given below;

  1. Please clarify your devices ITO/PANI/ITO, ITO/ZnO/ITO, ITO/ZnO/PANI/ITO, ITO/ZnO/PANI/ZnO/ITO, …etc.

Answer:Thank you for your professional feedback. Based on your suggestion, we have further detailed the schematic diagrams of ZnO, PANI, ZnO/PANI/ZnO heterostructures in Figure 1. We sincerely hope that the answer can meet your requirements. Thank you again for your professional feedback.

Original:

Figure 1. Schematic diagram of experimental assembly of ZnO/PANI/ZnO heterostructure .

Revised:

Figure 1. Schematic diagram of experimental assembly and electrochemical testing assembly of ZnO, ZnO/PANI, ZnO/PANI/ZnO heterostructures

  1. A bare PANI should be prepared for photodetector application. A comparison between bare ZnO, bare PANI, ZnO/PANI and ZnO/PANI/ZnO will be valuable.

 Answer:Thank you for your comments.Thank you very much for your guidance. In this work, we conducted photocurrent testing on all assembled UV optoelectronic components. We found that when pure PANI switches on and off the light, there is basically no change in photocurrent. The reason may be that PANI has a weak light response and is not very sensitive to ultraviolet light.We sincerely hope that the answers would meet your requirements. Thanks again for your professional comments.

  1. The authors coated PANI on ZnO nanosheet by using spin coating method as they mentioned in experimental section. But they wrote in Results and Discussion section the PANI in the form of nanowire. Is it right? Please give details of nanowire dimensions with SEM or FESEM images.

 Answer:Thank you for your comments.In the experiment, we prepared PANI nanowires and coated them on ZnO nanosheets through spin coating. The diameter of PANI nanowires prepared is approximately  100nm. The SEM image is shown in the  Figure 3(b).

Original:

 Figure 3 showed SEM images of ZnO nanosheets and ZnO/PANI nanocomposites on ITO substrates. Figure 3 (a-c) was a top SEM images of the ZnO nanosheets arrays, the magnification of the ZnO nanosheets arrays and the ZnO/PANI nanocomposite, respectively. And Figure 3(d) was a cross-sectional SEM image of the ZnO/PANI nanocomposite. In Figure 3 (a, b), it can be seen that the substrate was covered with many vertically arranged nanosheets, whose diameter and thickness were almost uniform. The prepared ZnO nanosheet arrays showed prefect growth on the surface of ITO conductive glass, and it was evenly and tightly distributed on the conductive surface of the ITO substrate. It can be seen that the method eliminated the tedious process of preparing the seed layer and did not affect its growth. The average diameter of ZnO nanosheets are about 4 μm, and its thickness was below 100 nm. We can also observe that the surface of these two-dimensional hexagonal nanosheets was smooth, and the nanosheets covered the entire ITO substrate. Figure 3(c) showed that the average diameter of PANI NWs was about 90 nm, and ZnO nanosheets were in complete contact with PANI NWs. It contributed to the formation of a p-n heterojunction between n-type ZnO and p-type PANI, which can improve the sensitivity of the sensor and beneficial for the contact between nanocomposite and ultraviolet light and the transfer of photogenerated electrons [41].

Figure 3. SEM images of ZnO nanosheet array (a, b) and ZnO/PANI nanocomposite structure (c, d).

Revised:

Figure 3 shows the SEM images of ZnO nanosheets and ZnO/PANI nanocomposites on ITO substrates. Figures 3 (a-c) show the top SEM images of ZnO nanosheet arrays, PANI nanowires, and the magnification of ZnO/PANI nanostructures, respectively. The size of PANI nanowires is approximately 80 nm. Figure 3 (d) shows the cross-sectional SEM image of ZnO/PANI nanocomposites. In Figure 3 (a), it can be seen that the substrate is covered with many vertically arranged nanosheets, with almost uniform diameter and thickness. The prepared ZnO nanosheet array grows well on the surface of ITO conductive glass and is evenly and tightly distributed on the conductive surface of ITO substrate. It can be seen that this method eliminates the tedious process of preparing the seed layer and does not affect its growth. The average diameter of ZnO nanosheets is about 4 μ m. The thickness is below 100nm. We can also observe that the surface of these two-dimensional hexagonal nanosheets is smooth, and the nanosheets cover the entire ITO substrate. Figure 3 (c, d) shows complete contact between ZnO nanosheets and PANI nanowires. It helps to form p-n heterojunctions between n-type ZnO and p-type PANI, which can improve the sensitivity of the sensor and facilitate the contact between nanocomposites and ultraviolet light, as well as the transfer of photogenerated electrons [41]

Figure 3.  SEM images of ZnO nanosheet array (a),PANI nanowires (b) and ZnO/PANI nanocomposite structure (c, d).

  1. The authors should make it clear in the Experimental method section which fabrication conditions of the ZnO nanosheet structures were investigated. Figure 2 shows the SEM images for ZnO nanosheets prepared under different conditions.

  Answer:Thank you very much for your reply. After our experiments, we found that temperature, time, and voltage have the greatest impact on the morphology and structure of ZnO nanosheets.Therefore, we control the morphology of ZnO by controlling these three factors. And three most representative morphologies were selected for comparison.  After comparison, it was found that the prepared ZnO nanosheets have the best structural morphology after reacting at -1.1V and 50 ℃ for 30 minutes. Therefore, the preparation of ZnO nanosheets is achieved by electrolysis of Zn (NO3) 2 KCl solution at -1.1V and 50 ℃ for 30 minutes and deposition on ITO conductive glass.

Original:

PANI NWs were prepared by chemical oxidation from aniline monomer. Firstly, dissolve 800 μL of aniline in 100 mL of 1 mol/L HCL solution and stir. Then 5 mL of 0.1 mol/L Ammonium persulfate (APS) solution was dissolved in a mixture of aniline and hydrochloric acid solution with stirring. The resulting mixture was plated in a freezer to keep the reaction temperature around 0-5 ℃ until the reaction was finally completed. Then, the obtained polyaniline solution was taken out, centrifuged and concentrated, followed by washing with anhydrous ethanol, repeated 2-4 times to prevent residual hydrochloric acid from corroding the ZnO nanosheet arrays. It was then coated on the ZnO nanosheets by a low-speed spin-coating method and dried in air. Finally, a ZnO NSs/PANI NWs heterostructure array was obtained. Then, the two samples were plated face to face and clamped them to obtain a double-layer ultraviolet photodetector. Figure 1 showed the preparation of ZnO/PANI nanocomposites and the schematic diagram of device assembly.ZnO nanosheet arrays were prepared by electrochemical method, PANI nanowire arrays were prepared by chemical oxidation method at low temperature, and composite materials were assembled by spin coating method to obtain the required single and double layer devices. A series of morphology, structure, and photoelectric performance tests were conducted on them.

Revised:

PANI NWs were prepared by chemical oxidation from aniline monomer. Firstly, dissolve 800 μL of aniline in 100 mL of 1 mol/L HCL solution and stir. Then 5 mL of 0.1 mol/L Ammonium persulfate (APS) solution was dissolved in a mixture of aniline and hydrochloric acid solution with stirring. The resulting mixture was plated in a freezer to keep the reaction temperature around 0-5 ℃ until the reaction was finally completed. Then, the obtained polyaniline solution was taken out, centrifuged and concentrated, followed by washing with anhydrous ethanol, repeated 2-4 times to prevent residual hydrochloric acid from corroding the ZnO nanosheet arrays. It was then coated on the ZnO nanosheets by a low-speed spin-coating method and dried in air. Finally, a ZnO NSs/PANI NWs heterostructure array was obtained. Then, the two samples were plated face to face and clamped them to obtain a double-layer ultraviolet photodetector. Figure 1 showed the preparation of ZnO/PANI heterostructure and the schematic diagram of device assembly.

Our work is to prepare a ZnO based ultraviolet photodetector. Using a three electrode system and a KCl solution of Zn (NO3)2 as the electrolyte, ITO/ZnO was prepared by electrodepositing ZnO nanosheets onto ITO conductive glass at -1.1 V and 50 ℃ for 30 minutes. Afterwards, spin coat PANI onto ITO on the ZnO surface, which is ITO/ZnO/PANI. Prepare two identical ITO/ZnO/PANI, and assemble the PANI face to face, namely ITO/ZnO/PANI/ITO.And a series of morphology, structure, and photoelectric performance tests were conducted on them.

  1. In this study, it should be made clear whether you have created a ZnO/PANI nanocomposite or a heterostructure. In the text, the term composite is used in some places and heterostructure in others.

 Answer:Thank you for your criticism and correction.We have created a p-n heterostructure ultraviolet optoelectronic device composed of p-type PANI nanowires and n-type ZnO nanosheets. We will unify this writing in the manuscript.

  1. The authors mentioned about Linear sweep voltammetry and AC impedance in the Experimental section. But they did not give any results.

 Answer:Thank you very much for your criticism and correction. I'm sorry that due to my mistake, I confused the Linear sweep voltammetry and AC impedance methods used in other work with this one, and these two methods were not used in this work. I'm very sorry for the inconvenience caused to your reading.

  1. In my opinion it is not necessary to define analyze or characterization methods in Experimental section such as “The X-ray powder diffractometer is mainly used to study and identify the composition and atomic level structure of substances and materials. It analyzes the composition, 147 particle size, crystallinity, etc. of nanoparticles by comparing them with the diffraction patterns in the powder diffraction file (PDF) database. From the powder diffraction spectrum, physical quantities such as diffraction peak position, intensity, and peak shape can be directly obtained.”

 Answer:Thank you very much for your guidance. I'm sorry that these introductions have caused obstacles to your reading. My intention was to introduce the experimental instruments used, but as you said, this is indeed too verbose and unnecessary. I will delete it in the manuscript.

Original:

2.4 Structural characterizations of materials

2.4.1 Field emission scanning electron microscope,FESEM

The working principle of field emission scanning electron microscope is to irradiate the sample with an accelerated high-energy electron beam in a high vacuum state, and the incident electron beam interacts with the sample to generate various signals. By detecting various signals with different detectors, various information about the sample can be obtained.

This project used JSM-7001F field emission scanning electron microscopy from JEOL Company in Japan. Cut the sample into appropriate sizes, place it on a metal sample table, spray gold for 80 seconds, accelerate the voltage to 15 kV, and observe the morphology of the sample after testing.

2.4.2 X-ray powder diffraction,XRD

The X-ray powder diffractometer is mainly used to study and identify the composition and atomic level structure of substances and materials. It analyzes the composition, particle size, crystallinity, etc. of nanoparticles by comparing them with the diffraction patterns in the powder diffraction file (PDF) database. From the powder diffraction spectrum, physical quantities such as diffraction peak position, intensity, and peak shape can be directly obtained.

This project used a PW3040/60 X-ray powder diffractometer from PANalytical in the Netherlands, with a Cu Ka target. The X-ray wavelength is 0.15406 nm, the scanning speed is 2 °/min, the step width is 0.02 °, the test current is 40 mA, the voltage is 40 kV, and the scanning angle is 10-80 °.

2.4.3 UV-Vis Spectrophotometer

The UV visible absorption spectrum of molecules is generated by the electronic energy level transition of certain functional groups in the molecule after absorbing UV visible radiation. Due to the different molecular, atomic, and molecular spatial structures of various substances, their absorption of light energy is also different. Therefore, each substance has its own unique and fixed absorption spectrum curve, and the content of the substance can be judged or measured based on the absorbance at certain characteristic wavelengths on the absorption spectrum.

This project used a UV-2450 ultraviolet visible spectrophotometer from Shimadzu Company in Japan.

2.4.4 Linear sweep voltammetry

Linear sweep voltammetry is an electrochemical experimental technique that applies a linearly changing voltage to the electrode and records the electrolytic current on the working electrode. The curve of the recorded current changing with the electrode potential is called a linear sweep voltammogram.Generally speaking, the voltage scanning range can be divided into two categories: small amplitude and large amplitude. The small voltage range is usually within 5-10 mV and is mainly used for measuring transition resistance and double layer capacitance. A wide range of voltage is commonly used for measuring electrode reaction parameters, determining the reversibility of electrode processes, controlling steps, and studying reaction mechanisms.

This project used CHI660D electrochemical workstation from Shanghai Chenhua Technology Co., Ltd. to perform linear sweep voltammetry testing on the device. The initial potential is -1.2 V, the termination potential is 1.2 V, the scanning speed is 0.1 V/s, and the sampling interval is 0.01 V.

2.4.5 Current time curve

The current time curve is a method of recording the variation of electrode current with time under a large step potential applied to the working electrode on a stationary electrode and in an unstirred solution (The potential that jumps from a potential without Faraday reaction to the surface active component of the electrode effectively approaches zero).

This project used CHI660D electrochemical workstation from Shanghai Chenhua Technology Co., Ltd. to test the current time curve of the device, with an external bias of 0 and a sampling interval of 0.1 seconds.

2.4.6 AC impedance

AC impedance method refers to the method of controlling the current (or potential) passing through an electrochemical system as a small amplitude sinusoidal AC signal, while measuring the corresponding system potential (or current) over time, or directly measuring the AC impedance (or admittance) of the system, analyzing the reaction mechanism of the electrochemical system and calculating the relevant parameters of the system. It has the advantages of low system interference, providing information on interface states and processes from multiple angles, and reliable results.

This project used the CHI660D electrochemical workstation of Shanghai Chenhua Technology Co., Ltd. to test the AC impedance of the device. The initial level is set through the open circuit potential time curve, with a frequency range of 106~0.1 Hz.

Revised:

2.4 Structural characterizations of materials

2.4.1 Field emission scanning electron microscope,FESEM

This project used JSM-7001F field emission scanning electron microscopy from JEOL Company in Japan. Cut the sample into appropriate sizes, place it on a metal sample table, spray gold for 80 seconds, accelerate the voltage to 15 kV, and observe the morphology of the sample after testing.

2.4.2 X-ray powder diffraction,XRD

This project used a PW3040/60 X-ray powder diffractometer from PANalytical in the Netherlands, with a Cu Ka target. The X-ray wavelength is 0.15406 nm, the scanning speed is 2 °/min, the step width is 0.02 °, the test current is 40 mA, the voltage is 40 kV, and the scanning angle is 10-80 °.

2.4.3 UV-Vis Spectrophotometer

This project used a UV-2450 ultraviolet visible spectrophotometer from Shimadzu Company in Japan.The light source is a deuterium lamp and a tungsten lamp, with a set ultraviolet wavelength range of 200-800 nm.Conduct UV response experiments on photodetectors using CHI660D electrochemical workstation. The ultraviolet source is provided by a portable ultraviolet lamp (ENF-280C, USA).The light intensity is 310 μW/cm2.

2.4.4 Current time curve

The current time curve is a method of recording the variation of electrode current with time under a large step potential applied to the working electrode on a stationary electrode and in an unstirred solution (The potential that jumps from a potential without Faraday reaction to the surface active component of the electrode effectively approaches zero).

This project used CHI660D electrochemical workstation from Shanghai Chenhua Technology Co., Ltd. to test the current time curve of the device, with an external bias of 0 and a sampling interval of 0.1 seconds.

  1. The authors have repeatedly used the phrase “This project used” in Experimental section. Why?

 Answer:Thank you very much for your criticism. I am very sorry for the serious colloquialism during the writing process of the article, which caused inconvenience to your reading. I will proofread the article and make changes.

Reviewer 3 Report

Comments and Suggestions for Authors

This article discussed the preparation of ZnO/PANI/ZnO sandwich-like structure and its potential application to ultraviolet photodetector. The authors demonstrated significant UV current improvement of such structure comparing with traditional ZnO devices, which was attributed to the p-n junction formed at ZnO/PANI interfaces. According to my opinion this work can be published on Polymers if the following issues can be addressed:

1.      Figure 2

Authors compared ZnO nanosheets prepared at different time, temperature, and voltage. Could the authors describe more details on the standard of such comparison? Was there a quantitative method of such comparison?

2.      Line 254

Authors calculated the average diameter of the PANI NWs. How was the value calculated? Could this method be used for other sample comparison?

3.      Figure 4

Did the stars indicate peaks of ITO substrate? A legend of the peaks might be needed.

4.      Figure 6

In Figure 6. (b) and (c), the I-V curves of the ZnO/PANI showed different shape to ZnO and ZnO/PANI/ZnO samples. The explanation of such difference was missing.

Comments on the Quality of English Language

Minor revision needed

Author Response

Reviewer #3:

This article discussed the preparation of ZnO/PANI/ZnO sandwich-like structure and its potential application to ultraviolet photodetector. The authors demonstrated significant UV current improvement of such structure comparing with traditional ZnO devices, which was attributed to the p-n junction formed at ZnO/PANI interfaces. According to my opinion this work can be published on Polymers if the following   issues can be addressed:

  1. Figure 2

Authors compared ZnO nanosheets prepared at different time, temperature, and voltage. Could the authors describe more details on the standard of such comparison? Was there a quantitative method of such comparison?Thank you 

 Answer:Thank you very much for your guidance. I am glad to receive your modification suggestions. Through experiments, we found that temperature, time, and voltage have the greatest impact on the morphology of ZnO nanosheets. Therefore, we conducted extensive experimental comparisons of these three factors and selected the three most representative ones. When the reaction time is 10 minutes, the growth of ZnO nanosheets is disordered. When the reaction time is 60 minutes, the growth layer of zinc oxide nanosheets is too thick; When the reaction temperature is set to 30 ℃, ZnO nanosheets will not grow and the layer is too thin at 40 ℃; When the voltage is set to -0.7V, ZnO nanosheets will not grow, but when the voltage is set to -1.5V, they severely aggregate and appear blocky. Therefore, after comprehensive comparison and consideration, we chose to react at 50 ℃ and -1.5V for 30 minutes to prepare ZnO nanosheets.We hope that our explanation can be accepted by you and thank you again for your very professional and meticulous comments.

  1. Line 254

Authors calculated the average diameter of the PANI NWs. How was the value calculated? Could this method be used for other sample comparison?

  Answer:Thank you very much for your feedback.I made a rough estimate of it through SEM images.

Original:

 Figure 3 showed SEM images of ZnO nanosheets and ZnO/PANI nanocomposites on ITO substrates. Figure 3 (a-c) was a top SEM images of the ZnO nanosheets arrays, the magnification of the ZnO nanosheets arrays and the ZnO/PANI nanocomposite, respectively. And Figure 3(d) was a cross-sectional SEM image of the ZnO/PANI nanocomposite. In Figure 3 (a, b), it can be seen that the substrate was covered with many vertically arranged nanosheets, whose diameter and thickness were almost uniform. The prepared ZnO nanosheet arrays showed prefect growth on the surface of ITO conductive glass, and it was evenly and tightly distributed on the conductive surface of the ITO substrate. It can be seen that the method eliminated the tedious process of preparing the seed layer and did not affect its growth. The average diameter of ZnO nanosheets are about 4 μm, and its thickness was below 100 nm. We can also observe that the surface of these two-dimensional hexagonal nanosheets was smooth, and the nanosheets covered the entire ITO substrate. Figure 3(c) showed that the average diameter of PANI NWs was about 90 nm, and ZnO nanosheets were in complete contact with PANI NWs. It contributed to the formation of a p-n heterojunction between n-type ZnO and p-type PANI, which can improve the sensitivity of the sensor and beneficial for the contact between nanocomposite and ultraviolet light and the transfer of photogenerated electrons [41].

Figure 3. SEM images of ZnO nanosheet array (a, b) and ZnO/PANI nanocomposite structure (c, d).

Revised:

Figure 3 shows the SEM images of ZnO nanosheets and ZnO/PANI nanocomposites on ITO substrates. Figures 3 (a-c) show the top SEM images of ZnO nanosheet arrays, PANI nanowires, and the magnification of ZnO/PANI nanostructures, respectively. The size of PANI nanowires is approximately 100nm. Figure 3 (d) shows the cross-sectional SEM image of ZnO/PANI nanocomposites. In Figure 3 (a), it can be seen that the substrate is covered with many vertically arranged nanosheets, with almost uniform diameter and thickness. The prepared ZnO nanosheet array grows well on the surface of ITO conductive glass and is evenly and tightly distributed on the conductive surface of ITO substrate. It can be seen that this method eliminates the tedious process of preparing the seed layer and does not affect its growth. The average diameter of ZnO nanosheets is about 4 μ m. The thickness is below 100nm. We can also observe that the surface of these two-dimensional hexagonal nanosheets is smooth, and the nanosheets cover the entire ITO substrate. Figure 3 (c, d) shows complete contact between ZnO nanosheets and PANI nanowires. It helps to form p-n heterojunctions between n-type ZnO and p-type PANI, which can improve the sensitivity of the sensor and facilitate the contact between nanocomposites and ultraviolet light, as well as the transfer of photogenerated electrons [41]

Figure 3.  SEM images of ZnO nanosheet array (a),PANI nanowires (b) and ZnO/PANI nanocomposite structure (c, d).

  1. Figure 4

Did the stars indicate peaks of ITO substrate? A legend of the peaks might be needed.

  Answer:Thank you very much for your guidance and I’m pleased to receive your modification suggestions.The peak marked with stars is the base peak of ITO conductive glass. I will explain it in the text. I apologize for forgetting to explain it in the manuscript, which caused you reading difficulties.

Original:

The XRD pattern of ZnO nanosheets was shown in Figure 4. All the diffraction peaks can point to a hexagonal fibrillar zincite structure. Except for the diffraction peaks of ITO substrate, no characteristic peaks of other impurities were detected, which indicated that only single-phase ZnO samples have been formed. Sharp diffraction peaks indicated good crystallinity of the synthesized ZnO nanosheets [38].

Revised:

The XRD pattern of ZnO nanosheets was shown in Figure 4. All the diffraction peaks can point to a hexagonal fibrillar zincite structure. The peak marked with stars is the base peak of ITO conductive glass.Except for the diffraction peaks of ITO substrate, no characteristic peaks of other impurities were detected, which indicated that only single-phase ZnO samples have been formed. Sharp diffraction peaks indicated good crystallinity of the synthesized ZnO nanosheets [38].

  1. Figure 6

In Figure 6. (b) and (c), the I-V curves of the ZnO/PANI showed different shape to ZnO and ZnO/PANI/ZnO samples. The explanation of such difference was missing.

Answer:Thank you for your  comments. ZnO itself has defects. When light is added, the current instantly increases, indicating that the increased current is generated due to the introduction of light. However, there will be a sharp peak when the current increases, possibly due to the rapid recombination of electron holes, resulting in a sharp decrease in photocurrent and the appearance of a sharp peak. The addition of PANI has to some extent compensated for this defect, causing the current to slowly rise. And ZnO/PANI/ZnO also experienced a phenomenon of sudden current drop, which may be due to the large defects in ZnO that PANI cannot fully compensate for.We sincerely hope that the answers would meet your requirements. Thanks again for your professional comments.

Original:

Figure 6(c) showed the photocurrent curves of single-layer ZnO NSs/PANI NWs nanocomposites and ZnO NSs/PANI NWs/ZnO NSs sandwich structure devices, with a bias voltage of 0 V and a UV wavelength of 365 nm for irradiation. Compared with single-layer ZnO NSs/PANI NWs nanocomposites, devices based on ZnO NSs/PANI NWs/ZnO NSs sandwich structure have significantly improved photosensitivity and photocurrent. In addition, compared to traditional double-layer structured nanocomposites, ZnO/PANI/ZnO sandwich structures exhibit greater photocurrent and high experimental repeatability. It can be seen that ZnO/PANI/ZnO sandwich structure devices have higher photoelectric conversion efficiency, which also provides new ideas for improving the performance of various aspects of ultraviolet photoelectric devices.Therefore, our UV photodetectors should be based on the ZnO NSs/PANI NWs/ZnO NSs sandwich structure. The I-t curves of ZnO NSs/PANI NWs/ZnO NSs devices under UV irradiation at wavelengths of 254 nm, 312 nm, and 365 nm are shown in Figure 6 (d). The results indicated that the photocurrent generated by ZnO/PANI/ZnO ultraviolet detectors under different wavelengths of ultraviolet light irradiation gradually increases with the increase of ultraviolet wavelength, and the experimental results were stable and have good repeatability.

Revised:

Figure 6(c) showed the photocurrent curves of single-layer ZnO NSs/PANI NWs nanocomposites and ZnO NSs/PANI NWs/ZnO NSs sandwich structure devices, with a bias voltage of 0 V and a UV wavelength of 365 nm for irradiation. Compared with single-layer ZnO NSs/PANI NWs nanocomposites, devices based on ZnO NSs/PANI NWs/ZnO NSs sandwich structure have significantly improved photosensitivity and photocurrent.In addition, as shown in Figure 6. (b) And (c), the I-V curves of ZnO/PANI show different shapes from ZnO and ZnO/PANI/ZnO samples. ZnO itself has defects. When light is added, the current immediately increases, indicating that the increased current is generated due to the introduction of light. However, as the current increases, sharp peaks appear, which may be due to the rapid recombination of electron holes, leading to a sharp decrease in photocurrent and the appearance of sharp peaks. The addition of PANI has to some extent compensated for this defect, leading to a slow increase in current. ZnO/PANI/ZnO also experienced a sudden drop in current, which may be due to PANI's inability to fully compensate for the large defects in ZnO. But it also can be seen that ZnO/PANI/ZnO sandwich structure devices have higher photoelectric conversion efficiency, which also provides new ideas for improving the performance of various aspects of ultraviolet photoelectric devices.

In addition, by comparing the switching ratio, responsiveness, and rise/fall time of photocurrent devices, as shown in Chart 1, it can be found that the switching ratio and responsiveness of ZnO/PANI/ZnO heterostructures are greater than those of ZnO and ZnO/PANI, and the time required for the current peak to rise or decay by 80% is less than that of ZnO and ZnO/PANI. This gives us reason to believe that the photoelectric performance of ZnO/PANI/ZnO heterostructure optoelectronic devices is the best.

Therefore, our UV photodetectors should be based on the ZnO NSs/PANI NWs/ZnO NSs sandwich structure. The I-t curves of ZnO NSs/PANI NWs/ZnO NSs devices under UV irradiation at wavelengths of 254 nm, 312 nm, and 365 nm are shown in Figure 6 (d). The results indicated that the photocurrent generated by ZnO/PANI/ZnO ultraviolet detectors under different wavelengths of ultraviolet light irradiation gradually increases with the increase of ultraviolet wavelength, and the experimental results were stable and have good repeatability.

Round 2

Reviewer 1 Report

Comments and Suggestions for Authors

Authors have replied to our queries well and incorporated the corrections in the Manuscript. Therefore,  this Manuscript may be published in 'POLYMER JOURNAL.'

Author Response

Thanks

Reviewer 2 Report

Comments and Suggestions for Authors

The revisions are mostly successful, but some minor revisions and editing are suggested:

1.     As I see from SEM images, PANI is in form of nanoporous film. PANI nanowire structure is not seen.

2.     As you prepared ZnO/PANI p-n heterojuction, please use the word “heterostructure” in the whole text, not composite.

Author Response

Reviewer #1:

The revisions are mostly successful, but some minor revisions and editing are suggested:

  1. As I see from SEM images, PANI is in form of nanoporous film. PANI nanowire structure is not seen.

Answer:Thank you very much for your guidance. I am pleased to receive your suggestions for modifications. Perhaps due to the issue with the SEM machine model, the SEM image we captured is not very clear and the structure of PANI is similar to that of nanoporous films, but its own structure is still composed of nanowires. When naming, we referred to the work of others and found that others would name this structure PANI nanowires or nanorods. The problem you mentioned does indeed exist, so we have changed the PANI nanowire structure mentioned in the original text to PANI nanoporous membrane to make the article more rigorous.We have revised the expressions and sincerely hope that the answers would meet your requirements. Thanks again for your professional comments.

  1. As you prepared ZnO/PANI p-n heterojuction, please use the word “heterostructure” in the whole text, not composite.

Answer:Thank you for your professional comments. We apologize for any inconvenience caused to your reading due to non-standard writing. We will make changes and corrections to this in the manuscript. Thanks again for your professional comments.